# New Insights into Alpine *Cortinariaceae* (Basidiomycota): Three New Species, Two Type Revisions, and a New Record for the Alpine Zone

**DOI:** 10.3390/jof9090942

**Published:** 2023-09-18

**Authors:** Jean-Michel Bellanger, François Armada, Alessandro Fellin, Pierre-Arthur Moreau

**Affiliations:** 1CEFE, CNRS, Univ. Montpellier, EPHE, IRD, INSERM, 1919, Route de Mende, CEDEX 5, F-34293 Montpellier, France; 2203, Montée Saint-Mamert-le-Haut, F-38138 Les Côtes-d’Arey, France; paco38@wanadoo.fr (F.A.); pierre-arthur.moreau@univ-lille.fr (P.-A.M.); 3Via G. Canestrini 10/B, I-38028 Novella, TN, Italy; fellin6384@yahoo.it; 4Laboratoire de Génie Civil et Géo-Environnement (ULR 4515-LGCgE), University Lille, F-59000 Lille, France

**Keywords:** alpine fungi, Cortinarius subgen. *Telamonia*, *Flexipedes*, *Castanei*, *Saniosi*, *Verni*, *Thaxterogaster*

## Abstract

Thirty-one alpine species of *Cortinarius* (Agaricales, Cortinariaceae) were described from the alpine zone of the Alps during the second half of the XX century, by the Swiss mycologist Jules Favre, and by the French mycologists Denise Lamoure and Marcel Bon. Notoriously difficult to identify by macro- and microscopical characters, most of these species, which belong to subgen. *Telamonia,* have been thoroughly revised in global publications based on type sequencing. Recent surveys in the alpine areas of France (Savoie) and Italy (Lombardy), as well as the sequencing of D. Lamoure’s collections, identified three new species that are here described and illustrated: *C. dryadophilus* in sect. *Castanei*, *C. infidus* in sect. *Verni*, and *C. saniosopygmaeus* in sect. *Saniosi*. The holotypes of *C. caesionigrellus* Lamoure and *C. paleifer* var. *brachyspermus* Lamoure could be sequenced. A recent collection of the former is described and illustrated here for the first time, and based on available data, the latter name is recombined as *Cortinarius flexipes* var. *brachyspermus* comb. nov. Lastly, *C. argenteolilacinus* var. *dovrensis* is reported from the alpine zone for the first time and a new combination, *Thaxterogaster dovrensis* comb. & stat. nov. is introduced in the present work.

## 1. Introduction

The genus *Cortinarius* is the most species-rich fungal species, even after its revision and emendation by Liimatainen et al. [1]. Its subgenus *Telamonia* itself represents the largest currently recognized part of the genus with 184 species and 80 sections identified by Liimatainen et al. [2]. In the Mediterranean, temperate, and arctic–alpine zones of the Northern Hemisphere, *Telamonia* species are dominant in ectomycorrhizal communities, although their taxonomy remains one of the least resolved according to the above-mentioned molecular works. Alpine communities associated with dwarf *Salix* species, *Dryas octopetala,* and *Polygonum viviparum* are especially dominated by small *Telamonia* species.

Jules Favre [3,4] was the first meticulous observer of these Lilliputian communities in the hardly accessible slopes of the Swiss National Park, from where he reported 26 species of *Cortinarius*, 19 of them new to science. After Favre’s era, 25 years of intensive prospections took place, carried out by Robert Kühner and Denise Lamoure in the French alpine zone of the National Park of Vanoise [5,6]. A total of 38 species of *Cortinarius* were recorded by them, 10 newly described [7,8], and 1 previously described by another famous promotor of alpine mycology, Marcel Bon [9]. Finally, Bon keyed out 56 species for the European alpine zone [10], 38 of them in the currently delimited subgen. *Telamonia*.

With the exception of *Cortinarius tatrensis* Fellner & Landa described from the Tatras Mountains [11], and in spite of intensive forays into many alpine areas throughout Europe, no new *Cortinarius* species were subsequently published from these ecosystems, indicating that Favre’s [3,12], Lamoure’s [7,8,13], and Bon’s [10] taxonomies remained the reference works for the identification of alpine *Cortinarius* in Europe for more than three decades.

The DNA-based revisions of *Telamonia* simultaneously published by Liimatainen et al. [2] and, specifically focusing on alpine taxa, by Kokkonen [14], have clearly highlighted the need to revise the entire traditionally accepted taxonomy for these small fungi. The sequencing of the type collections was a key step for this ambitious challenge. Liimatainen et al. [2] published 11 ITS type sequences from Favre’s alpine species of *Telamonia*, and Kokkonen [14] published 6 more from Favre’s species and 3 from Lamoure’s taxa. While most species have thus been elucidated (unlocking the identification of many arctic–alpine collections and environmental samplings), those which could not be sequenced for various reasons remain enigmatic.

A series of prospections in the Vanoise region (Savoie, France) close to Kühner and Lamoure’s areas by F. A. and P.-A. M. between 2010 and 2019, and privileged access to D. Lamoure’s herbarium (LY) in 2020, yielded important alpine material that could be sequenced and compared to the data already published [2,14]. Most of the collections could be attributed to taxa already reported from Vanoise [6] and confirmed here, but some of them were of special interest and are the object of the present article. Introduced here are *Cortinarius infidus* sp. nov., a sibling species of *C. inops* J. Favre (sect. *Verni*), *C. dryadophilus* sp. nov. (sect. *Castanei*), and *C. saniosopygmaeus* sp. nov. (sect. *Saniosi*). A sequence obtained from the holotype of *C. caesionigrellus* Lamoure reveals the phylogenetic identity of this taxon within the sect. *Flexipedes.* A recent sequenced collection of this species further allows its taxonomic update and illustration for the first time. In addition, the new combinations *Cortinarius flexipes* var. *brachyspermus* (Lamoure) comb. nov., based on type revision, and *Thaxterogaster dovrensis* (Brandrud) comb. & stat. nov., with a description of the first record from the alpine zone, are introduced here.

## 2. Materials and Methods

Studied material: Recent collections were carried out in Savoie, France, by F. A., and in Lombardy, Italy by A. F., all in alpine areas. Type and additional collections are kept in the herbarium of the University of Lyon, France (LY), and exceptions are mentioned in the “Material studied” parts in the Taxonomy section.

Morphological studies: fresh basidiomata were photographed in situ and the habitat, altitude, soil characteristics, and significant plants were noted. Detailed observations of macromorphological characters were performed on fresh, dehydrated, and rehydrated material, and compared to photographic material and field notes. Micro-anatomical studies were conducted on sections from the pileus, stipe, and lamellae, revived and mounted in 10% ammoniacal Congo red, or in 3–5% KOH. Spore measurements were taken from natural deposits on the stipe surface obtained from dry basidiomata, mounted in water or in Congo red, and measured using the freeware Piximètre 5.10 [15]. A minimum of 30 spores were measured from each basidiome and the Me (average length and width), Q (minimum and maximum length/width ratio), and Qm (average length/width ratio) were calculated. The spore measurements exclude the apiculus.

DNA extraction, amplification, and sequencing: DNA extraction and PCR amplification were conducted with the REDExtract-N-Amp^tm^ Plant PCR Kit (Sigma-Aldrich, St. Louis, MO, USA), following the manufacturer’s instructions. The internal transcribed spacers and 5.8S of nuc rDNA (ITS barcode) were amplified from each collection, with the primers ITS-1F, ITS-4b, ITS-4, ITS-2, and ITS-3 [16,17], as described in [18]. The amplicons were purified by Eurofins Genomics, Ebersberg, Germany, prior to sequencing. Raw sequence data were edited and assembled with Codon Code Aligner 4.1.1 (CodonCode Corp., Centerville, MA, USA), and deposited in GenBank under the accession numbers indicated in the Taxonomy section below (type material and additional material).

Phylogenetic analyses: The ITS sequence dataset analysed here was assembled by combining 19 newly generated sequences with 54 phylogenetically close published sequences from the *Cortinarius* subgen. *Telamonia*, identified by BLAST searches in GenBank and UNITE. Multiple sequence alignment was carried out with MUSCLE 3.7 [19] and edited manually to adjust some homologous indels. Maximum likelihood phylogenetic analysis of the aligned sequences was performed online at www.phylogeny.lirmm.fr [20] with PhyML 3.0 [21], using the GTR model of evolution. Branch support was assessed using the non-parametric, Shimodaira–Hasegawa, version of the approximate likelihood-ratio test (SH-aLRT), implemented in the latest release of PhyML and which ensures high accuracy when SH-aLRT > 0.81 [22]. The phylogenetic tree (depicted in Figure 1) was edited with Inkscape 0.91 (https://inkscape.org/fr/, (accessed on 20 September 2021)).

## 3. Results and Discussion

### 3.1. New Taxa in Cortinarius Sect. Castanei, Sect. Verni, and Sect. Saniosi

***Cortinarius dryadophilus*** Armada, Bellanger & P.-A. Moreau, **sp. nov**. MycoBank MB849751. (Figure 2A,B).

*Etymology*: Latin, that likes *Dryas* thickets, relative to the ecology of this species.

*Type materia*l: FRANCE, Savoie, Peisey-Nancroix, along the GR5 path on the way to the lake of La Plagne, 2050 m asl, among *Dryas octopetala*, 25 August 2018, F. Armada (holotype LY FA4330). GenBank ITS OR419975.

*Diagnosi*s: A date-brown or vinaceous-brown «*Decipientes*» with reddish tinges in the stipe evoking a small-spored *C. subtribulosus*, in *Dryas* thickets.

Pileus 15–28 mm in diameter, conical to conical-convex; surface vinaceous reddish-brown to dark date-brown at the centre, lighter towards the margin, hygrophanous, covered by a whitish fibrillose or araneous veil, abundant especially at the margin; margin slightly incurved, early fimbriate, whitish, appendiculated by thin remnants of cortina. Lamellae up to 4 mm wide, crowded, thin, adnate, rather irregular in length, beige to dark beige, and rusty brown when mature; edge entire, somewhat sinuous, subconcolorous to the faces or paler. Stipe 31–43 × 3–5 mm, slender, cylindrical fistulose on adults; surface distinctly blueish at the apex, below entirely covered by a whitish fibrillose-silky veil forming a small annuliform zone at the upper third, hiding a vinaceous reddish-brown background visible by detersion. Context moderately thick at the centre, thin elsewhere, and reddish-brown or vinaceous, lilacine at the stipe apex. Smell weak, pleasant, and slightly fruity or pastry-like. Taste mild. Spore print not obtained, cinnamon ochre on stipe surface. KOH reaction: bistre-brown on pileus surface and context.

Spores (7.5) 8–9.5 (10.5) × (5.25) 5.5–6 (6.5) µm, ellipsoid-pruniform, with a fairly marked hilar depression, strongly ornamented by pointed or echinulate-truncate warts up to 0.5 µm high, higher at the apex where they can reach 1 (1.25) µm. Basidia 28–46 × 8–10 µm, cylindrical to slightly clavate, tetrasporic, and clamped. Sterile marginal cells 4–8 µm wide, common, and basidioid; edge fertile. Pileipellis made of a thin layer of narrow hyphae 1–5 µm wide covering sausage-shaped hyphae up to 16 µm wide; strong yellow or dark yellow pigmentation, wall coating, sometimes finely incrusting the upper hyphae; clamped.

Habitat and distribution: So far only known from the type locality but most likely present, albeit overlooked or misidentified, in other ecologically similar areas.

*Notes*: We introduce *C. dryadophilus* as a new species here based on a single collection, because of its unique molecular signature and because its ecology may help identify it in the field. The ITS sequence of this species is unknown from the GenBank and UNITE public databases, but it clearly belongs to the sect. *Castanei* [2] (Figure 1). This lineage encompasses the contested neotype of *C. castaneus* (Bull.) Fr., that of *C. decipiens* (Pers.) Fr., several historical names that were sometimes merged into a *C. decipiens sensu lato* concept [23], and species more recently identified or validated by molecular tools, such as *C. cistocastaneus* Armada & Bellanger [24] and *C. rufomyrrheus* Eyssart., Sleiman & Bellanger [25]. As delineated in “Mission Impossible” [2], sect. *Castanei* consists of a strongly supported clade of mostly European species, associated with a handful of North American, more distantly related species, some yet to be named. Our new species nests in a basal position to the European clade and its inclusion in the latter clade further supports it (Figure 1). Its phylogenetic closest neighbour is *C. subodoratus* Bidaud, from which it differs by five nucleotide changes (three SNPs + two indels) [2].

Prior to DNA sequencing, we intended to relate *C. dryadophilus* to previously known alpine species [3,7,8,10]. *Cortinarius basiroseus* P.D. Orton, originally described from the calcicolous beech forest of Netley Park in southern England [26], was targeted because of apparently compatible colours and microscopic features. The sequencing of the holotype of *C. basiroseus*, successfully achieved during this work, evidenced the unexpected synonymy of this name (with nomenclatorial priority) with *C. rubrocinctus* Reumaux and *C. uraceoarmillatus* Bidaud, which together founded the sect. *Rubrocincti* [2] distant from sect. *Castanei* to which *C. dryadophilus* belongs.

Another red-stiped alpine *Cortinarius* sect. *Telamonia* is *C. tatrensis* [11]. A short sequence from the holotype of this species is available [2] and, although its phylogenetic placement remains uncertain, its conspecificity with the present species can be ruled out.

Morphologically, *C. dryadophilus* differs from *C. castaneus* (neotype) by larger spores (8–9.5 × 5.5–6 vs. 6.5–8 × 4.5–5.5 µm, respectively), lamella colour (beige or dark beige vs. bright reddish-ochre, respectively), and a different smell (fruity or pastry-like vs. slightly raphanoid to cedarwood, respectively). Another closely related species is *C. decipiens*, neotypified [27] by a collection from France under *Picea abies*, and that has since been widely observed from the plains to the Alps under various tree species. Its spores are of the same size as those of *C. dryadophilus* and only the colour of the lamellae, perhaps more often with violet tinges, may tell the two species apart. As a perfect lookalike of *C. decipiens*, *C. falsosus* Moën.-Locc. & Reumaux [28] displays the same risks of confusion with *C. dryadophilus*. With darker colours of both the pileus and lamellae, *C. fuscoflexipes* M.M. Moser & McKnight [29] is macroscopically more distinct from our species, but it produces similar spores. Despite being originally described from Yellowstone National Park in the USA, it was recently reported in Finland [14] and may thus co-occur in alpine localities with *C. dryadophilus*. The recently described *C. rufomyrrheus* [25] also displays real morphological similarities with *C. dryadophilus* but differs by smaller and narrower spores and a smell evoking burnt meat. Quite surprisingly, the species most resembling *C. dryadophilus* macroscopically does not belong to the sect. *Castanei*, but to sect. *Bombycini*. *Cortinarius subturibulosus* Kizlik & Trescol is indeed a perfect lookalike of the present species, with fortunately larger spores and usually distinctive smell of orange blossoms [30]; both may co-occur in the same *Dryas* patches, as was the case for our collection.

***Cortinarius infidus*** Armada, Bellanger & P.-A. Moreau, **sp. nov**. MycoBank MB849752. (Figure 3A–F).

*Etymology*: Latin, unfaithful, referring to the misleading taxonomic features of this species, and associated confusion with its sister species *C. inops*.

*Type materia*l: FRANCE, Savoie, Peisey-Nancroix, Lac Marlou, 2500 m asl, among *Salix herbacea*, *S. retusa,* and *Polygonum viviparum*, 21 August 2018, F. Armada (LY FA4276, holotype). GenBank ITS OR420012.

*Diagnosi*s: Similar to *Cortinarius inops* from which it differs by somewhat longer spores; otherwise characterized by a conspicuous white veil visible at least at the pileus margin, vinaceous-red tinges at least in the stipe flesh, spores reaching 10 µm, and a strict association with dwarf *Salix* carpets.

*Additional materia*l: FINLAND, Keski-Pohjanmaa, Kalajoki, Hiekkasärkät, near *Salix phylicifolia* and *Alnus* sp., sandy soil, 10 September 2004, K. Kokkonen 255/04 (not studied), GenBank ITS MN841140; Kokkola, Lohtaja, Vattajaniemi, beach, among S. repens and *Pinus sylvestris* seedlings, 11 September 2004, K. Kokkonen 265/04 (not studied), GenBank ITS MN841141; Pietarsaari, Fäboda, beach, near *Salix phylicifolia*, with *Alnus*, *Pinus,* and *Betula* spp. further away, 12 September 2004, K. Kokkonen 280/04 (not studied), GenBank ITS MN841143. FRANCE, Savoie, Peisey-Nancroix, Lac Marlou, 2500 m asl, among *S. herbacea*, 17 August 2010, F. Armada & E. Armada, FA 1734, GenBank ITS OR420007; ibid., among *Salix herbacea*, *S. retusa,* and *Polygonum viviparum*, 21 August 2018, F. Armada, FA 4281, GenBank ITS OR420013; ibid., Val-d’Isère, Col de l’Iseran, among *S. herbacea* (*S. retusa*, *S. reticulata*, *S. hastata,* and *P. viviparum* nearby), 2650 m asl, 23 August 2018, F. Armada, FA 4298, GenBank ITS OR420014. SWEDEN. Lapland, Lule Lappmark, Gällivare, Nieras, among *S. herbacea*, 30 August 2016, K. Kokkonen 1395/16 (not studied), GenBank ITS MN841144. SWITZERLAND, Graubünden. Scuol, Lai Sesvenna, among *S. herbacea*, 2660 m asl, 17 August 2017, K. Kokkonen 281/17 (not studied), GenBank ITS MN841142; ibid., 25 August 2019, K. Kokkonen 38/19 (not studied), GenBank ITS MN841150. ***Cortinarius inops*:** FRANCE, Savoie, Peisey-Nancroix, col de la Chal, 2500 m asl, among *S. retusa*, 25 August 2005, F. & E. Armada, FA 27, GenBank ITS OR420011; ibid., 13 August 2010, FA 1697, GenBank ITS OR420006; Val d’Isère, col de l’Iseran, 2650 m asl, among *S. herbacea*, *S. retusa,* and *S. reticulata* (*P. viviparum* and *S. hastata* nearby), 23 August 2018, F. Armada, FA 4310, GenBank ITS OR420015; ibid., 14 August 2019, FA 4874, GenBank ITS OR420016; Haute-Savoie, Les Contamines-Montjoie, col du Joly, 2000 m asl, among *S. reticulata*, *S. retusa,* and *Dryas octopetala*, 22 August 2010, F. & E. Armada, A. Bidaud & R. Fillion, FA 1767, GenBank ITS OR420008.

Pileus 4–33 mm in diameter, conical to conical-convex and remaining so, with an obtuse but pronounced umbo, sometimes acute; surface faintly viscid when wet, quickly dry, glabrous, entirely covered by a white fibrillose veil, abundant at the margin when young, on an almost black background, then reddish-brown to brownish, strongly hygrophanous, drying from the centre to reddish-brown, vinaceous-brown, or dark chestnut; margin incurved, sometimes crenulate or striate, incised on very old specimens, becoming fibrillose-fissurate at the end (like *Inocybe* spp.). Lamellae up to 5–6 mm wide, rather thick, broadly adnate, moderately crowded, light brown café-au-lait or beige, then early brown to rusty-brown; edges smooth, concolorous, or slightly paler than faces. Stipe 10–45 × 1.5–6 mm, slender, straight, or often curved at mid-height, distinctly clavate at base; surface entirely covered by an abundant whitish veil, forming or not a ring-like zone, on a vinaceous to reddish-brown or beige-brown ground visible by detersion; cortina white, loosening, early appressed. Context fairly thick at the centre, vinaceous, reddish-brown to brown when moist, and whitish and greyish when dry. Smell indistinct to slightly raphanoid or of cedar wood when cut. Taste mild. KOH reaction: brownish-black on the pileus surface and context. Spore print not obtained.

Spores (7) 7.5–10 (10.5) × (4.75) 5.25–5.5 (6.25) µm, yellowish-ochraceous in KOH, ellipsoid, pruniform to subamygdaliform, usually with strong ornamentation made of truncate warts up to 1.25 µm high, denser at the apex, with a visible apicular depression. Basidia 24–40 × 7.5–11 µm, tetrasporic. Sterile marginal cells, 5–18 µm wide, cylindric-clavate or paddle-like, sometimes difficult to see but always very abundant, at least locally, and septate. Pileipellis made of three layers of hyphae, reaching respectively 1.5–4 µm, up to 16 µm, and up to 32 µm wide, with coarse dark yellow encrusting pigment, sometimes forming broad patches. Clamps present at all septa.

*Habitat and distribution*: So far confirmed from alpine dwarf *Salix* microsylvae (*S. herbacea*, *S. retusa*, *S. reticulata*, and *S. hastata*) in the French and Swiss Alps above 2500 m asl and at lower elevations in northern Sweden, and on seashores of Finland, associated with *S. repens* or *S. phylicifolia*. Probably widespread in boreoarctic and alpine areas with *Salix* spp., but likely often confused with *C. inops*.

*Notes*: Identifying a member of the “*Cortinarius inops* complex” is not too difficult in the field because of their shared dark basidiomata, more or less abundant whitish veil, and frequent pinkish-lilac or vinaceous tinges of the flesh. However, it remains challenging to unambiguously resolve them solely on morphological bases. Consistently, the present species was first identified [14] as a “group 3” unnamed ITS and rpb2 lineage very close to, but phylogenetically distinct from, two lineages including the lectotype of *C. inops* J. Favre [12] and the holotype of *C. suberythrinus* Moën.-Locc., respectively. Concomitantly, published in the famous type study “Mission Impossible“ [2] was an ITS sequence of the holotype of *C. tenebricus* J. Favre, that belongs in sect. *Verni*, close to *C. suberythrinus*. The combined analysis of the two datasets (Figure 1) supports a complex of three cryptic or semi-cryptic species that includes *C. inops* (syn. *C. tenebricus*), *C. suberythrinus* (syn. *C. vernus* var. *nevadavernus* Suár.-Sant. & A. Ortega, ? syn. *C. erythrinus* Fr.), and *C. infidus* sp. nov. that was not rare in our prospected area of the French Alps. *C. infidus* differs from *C. inops* at the ITS locus by four nucleotide changes (2 SNPs + two indels) and at the rpb2 locus by 14 SNPs; it differs from *C. suberythrinus* by two nucleotide changes (1 SNP + one indel) only at the ITS locus but by 11 SNPs at the rpb2 locus [14].

Ecologically, *C. infidus* differs from *C. suberythrinus* by its arctic–alpine distribution, so far only recorded in association with *Salix* spp., whilst *C. suberythrinus* is more ubiquitous but not known to be associated with *Salix*. *Cortinarius infidus* and *C. inops* seem to share the same ecological niches and both were even observed once on the same spot (Col de l’Iseran, 23 August 2018). Such sympatric occurrence of two populations with fixed SNPs at two distinct loci, with no sign of gene flow (e.g., heteromorphisms at the variable sites) strongly supports the existence of a tight reproductive barrier between them and subsequently supports two distinct species. The analysis of our 10 collections belonging to this complex pointed spore length as a possible (subtle) diagnostic criterion to distinguish *C. infidus* and *C. inops* by non-molecular tools. Indeed, although the average length values are not much different between the two species (7.5–10 µm vs. 7–9,5 µm, respectively), we observed that the number of spores reaching 10–10.5 µm was significantly higher in *C. infidus*.

Other morphological features claimed by past authors to display taxonomic value within the complex, like the presence/absence of cheilocystidia [3], the pinkish colour of the flesh [6,8], or the smell of cedar wood [31], were not supported here as a diagnostic to separate the two species. Cheilocystidia may be overlooked if only a portion of the lamella is observed without scrutiny, and these sterile elements are more difficult to detect on exsiccata, as they are often collapsed. All our personal collections of *C. infidus* and *C. inops* displayed conspicuous cheilocystidia. The reported lack of these sterile elements in *C. tenebricus* by Favre himself and by Horak on the same material [3,12] illustrates the challenging detection of such a “key feature” emphasized for instance by Bon [32]. Cheilocystidia are also present in *C. oreobius* J. Favre [3,12], a morphologically and ecologically closely related species with slightly different spores, but whose phylogenetic identity has not yet been established. *Cortinarius pusillus* F.H. Møller might be a candidate for an earlier, but illegitimate name for *C. infidus*. This species described from the Faeroe Islands was considered for instance by Moser [33], but not by Bon [10], as conspecific with *C. inops*.

***Cortinarius saniosopygmaeus*** Armada, Bellanger & P.-A. Moreau, **sp. nov**. MycoBank MB849753. (Figure 4A,B)

*Etymology*: Latin, evoking sect. *Saniosi*, which this species belongs to, and *Cortinarius phaeopygmaeus*, which it resembles morphologically.

*Type material*: FRANCE, Savoie, Peisey-Nancroix, Col de la Chal, 2500 m asl, among *Salix herbacea*, 17 August 2010, E. Armada & F. Armada (LY FA1724, holotype). GenBank ITS OR420003.

*Diagnosis*: A perfect lookalike of the ubiquitous *C. saniosus* or of its alpine synonym *C. chrysomallus*, with larger spores reaching 11 × 7 µm and fruiting in *Salix herbacea* carpets.

Pileus 7–20 mm in diameter, conical-convex then convex-flattened, with a small obtuse but fairly pronounced umbo, attenuated on older specimens; surface dry in early stages, dark reddish-brown gradually darker at the centre, not striate even when moist, not distinctly hygrophanous, made slightly fibrillose by a sparse yellowish-ochre veil especially visible at the margin, and glabrous when old; margin incurved, incised at the end, rendered orange-yellow by the veil, then glabrous and almost concolorous to the rest of surface on adults. Lamellae up to 3.5 mm wide, somewhat ventricose, broadly adnate to irregularly uncinate, sometimes subdecurrent, subdistant (17–23 reaching the stipe, one [-two] short plural per lamella), thin, pale beige-orange then rusty-orange; edge entire, subconcolorous to the face. Stipe 10–19 × 1–3 mm, straight or curved, and never fistulose; surface rendered entirely fibrillous by a persistent yellow veil, often forming a distinct ring-like zone at the lower third, showing a brownish ground colour when handled. Context not very thick, reddish-brown to dirty yellowish-white in the pileus. Smell not detected. Taste mild. KOH reaction: black, immediate on the pileipellis and context. Spore print not obtained.

Spores 9–11 (11.5–12.25) × (6) 6.5–7 (7.5) µm, yellowish-ochraceous in KOH, ellipsoid to subamygdaliform, with a small apicular depression, moderately verrucose, more coarsely at the apex. Basidia 28–39 × 8.5–11.5 µm, cylindrical or narrowly clavate, and tetrasporic. Sterile marginal cells very variable, locally abundant to nearly absent, cylindrical or racket-shaped, septate, and 7–10 µm wide. Pileipellis and trichocutis made of a layer of erected hyphae 1–3 µm wide, covering broader hyphae reaching 8–10 µm wide, and coarsely incrusted by a dark yellow to reddish epiparietal pigment. Subpellis a layer of allantoid hyphae reaching 28 µm wide, slightly less pigmented or encrusted than in the pileipellis. Clamps present at all septa.

Habitat and distribution: So far only confirmed from alpine *Salix herbacea* thickets of Austria, France, and Switzerland.

*Additional material*: AUSTRIA, Ötztal, alpine *Salix herbacea* snowbed, 22 August 1990, E. & K. Bendiksen, EKB 90.0822 (as *C. chrysomallus*, not studied), GenBank ITS DQ102656. SWITZERLAND, Graubünden. Scuol, Lai Sesvenna, near *S. herbacea*, 2665 m asl, 25 August 2019, K. Kokkonen, KK 35/19 (not studied), GenBank ITS MN841233; ibid., Val Sesvenna, near *S. herbacea*, 2525 m asl, 14 August 2017, K. Kokkonen, KK 306/17 (not studied), 307/17 (not studied), GenBank ITS MN841232; ibid., 360/17 (not studied); ibid., 17 August 2017, KK 283/17 (not studied), 316/17 (not studied). ***Cortinarius saniosus*:** FRANCE, Savoie, Peisey-Nancroix, Lac Marlou, 2500 m asl, among *Salix herbacea*, 21 August 2008, F. Armada, E. Armada & N. Van Vooren, FA 1069, GenBank ITS OR420002; Bonneval-sur-Arc, col de l’Iseran, 2650 m asl, among *Salix retusa*, 23 August 2018, F. Armada, FA 4317, GenBank ITS OR420005. ***Cortinarius subsaniosus***: FRANCE, Savoie, Peisey-Nancroix, Lac Marlou, 2500 m asl, among *Salix herbacea*, 19 August 2010, F. Armada, E. Armada, A. Bidaud & E. Bidaud, FA 1751/AB 10-08-57, GenBank ITS OR420004.

*Notes*: The present lineage has been known for a while but has not been formally named so far. In a landmark taxonomic revision of the *C. saniosus* (Fr.) Fr. complex, Lindström et al. [34] identified a group of four sequences in a so-called “clade 2”, sister to *C. saniosus*. Three of these sequences were later shown [14,35] to belong to *C. subsaniosus* Liimat. & Niskanen, a perfect lookalike of *C. saniosus*, with just slightly larger spores and a possible narrower distribution in lowland or coastal sandy areas. The fourth sequence in clade 2, GenBank DQ102656, originated from an Austrian alpine collection with *Salix herbacea* and was initially identified (and confirmed by M. Moser) as *C. chrysomallus* Lamoure [36]. Six additional collections were cited from similar ecosystems in Switzerland, with ITS sequences identical to DQ102656 [14]. Despite having pointed out some morphological differences in comparison with *C. saniosus* and *C. subsaniosus*, Kokkonen [14] refrained from formally naming this *S. herbacea*-associated alpine *Saniosi*. The three species co-occurred in our prospecting areas with *Salix herbacea* in a higher alpine zone, and we could compare them in detail for this study.

*Cortinarius subsaniosus* has not been recorded in the alpine zone to date. Our above-listed collections of *C. subsaniosus* and of *C. saniosus* were determined in the field as *C. saniosus* or *C. chrysomallus*, which were both expected in alpine snowbeds according to the usual literature [7,8,9,10]. Collection FA 1724 (*C. saniosopygmaeus*) was thought to represent *C. phaeopygmaeus* J. Favre, because of slightly larger spores than the two previous *Saniosi* species. The latter name was considered one of the less-well-characterized and interpretable alpine *Cortinarius* species [10,12,36] till its lectotype [12] was successfully sequenced and revealed that *C. phaeopygmaeus* is a member of the sect. *Flexipedes* [37], excluding conspecificity with the present species.

Spore size may be a key feature to disentangle this species complex: FA1724 displayed spores reaching 11 × 7 µm, whilst our collections of *C. saniosus* displayed spores measuring (8) 8.5–10 × 5–6.5 µm in accordance with previous studies [14,34,35]. Our unique collection of *C. subsaniosus* displayed similar spores, measuring 7.5–10 (10.5) × 5.5–6.5 µm, thus smaller than those published in the protologue (9.5–11 × 6–7 µm, MV = 10.2 × 6.1 µm [35]) but in accordance with Kokkonen (8–10 × 5.5–6 µm, MV = 9.1 × 5.7 µm), suggesting some phenotypic plasticity within *C. subsaniosus*, already reported at the macroscopic level [14].

Although spores of more sequenced collections of the three species should be carefully measured to confirm their respective range of variations, we here define *C. saniosopygmaeus* sp. nov. as a strictly *Salix herbacea*-associated species with distinctly larger spores on average than other alpine *Saniosi*.

### 3.2. Type Revisions and Updates

***Cortinarius caesionigrellus*** Lamoure, Trav. Sci. Parc Natl. Vanoise 9: 82 (1978) (Figure 5).

*Material studied*: FRANCE, Savoie, Bonneval-sur-Arc, behind l’Ouille-des-Reys, 2650 m, among *Salix herbacea*, 22 August 1971, D. Lamoure (LY L. 71-51, holotype), GenBank ITS OR419961. ITALY*,* Lombardy, Sondrio, Valfurva, passo Gavia, 2550 m asl, in an alpine meadow of *Salix herbacea* and *Alchemilla pentaphyllea*, 46°21′25.5″ N 10°30′02.7″ E, 18 August 2021, A. Fellin (TR gmb 01247), GenBank ITS OR420018; ibid. 31 July 2018 (TR gmb 01248), not sequenced.

Pileus (6) 8–25 mm in diameter, hemispherical to convex-appressed, in young specimens also campanulate, with central umbo barely outlined; inflexed margin, finely incised; cuticle dry, fibrillose, with a tendency to fissure radially or more irregularly towards the centre, covered by white-greyish araneous veil remnants or small appressed scales in young specimens, more abundant and persistent towards the margin, blackish-brown when wet, chestnut-brown, and lighter brown towards the margin in dry weather. Lamellae adnate-emarginate, distant, broad, sinuous, finely veined on the faces, brownish but with conspicuous and uniform lilac-purple tinges; edge entire or weakly indented, paler, and greyish; interspersed with short lamellulae. Stipe 8–25 (30) mm, cylindrical, sometimes curved, attenuated or barely clavate at the base or enlarged in the middle, striated, fibrillose, and covered with scattered or irregularly arranged greyish-white veil remnants; ground colour brownish and subconcolorous to the pileus; stipe base and mycelial felt occasionally with bluish hues. Context concolorous or subconcolorous with the pileus, darker and sometimes with a bluish tinge towards the stipe base and in the cortex; slight smell of *Pelargonium*.

Spores (8.1–) 8.5–9.1–9.6 (–10.2) × (5.4–) 5.6–5.9–6.2 (–6.5) µm, Q = (1.3–) 1.44–1.54–1.65 (–1.78), yellowish-ochraceous in KOH, dextrinoid, broadly ellipsoid to subovoid with rather variable profile, and finely verrucose; ornamentation consisting of low punctiform or trapezoidal warts, often larger and coalescent at the apex. Basidia 28–42 × 8–11 µm, claviform, hyaline or brownish in KOH, and mostly tetrasporic but with the presence of sporadic trisporic and bisporic basidia. Sterile marginal cells trivial, cylindric-clavate, basidioliform, interspersed between basidia, hyaline, and 13–28 × 4–8 µm. Pileipellis consisting of a thin layer of hyaline or at most yellowish hyphae 2–4 µm wide (suprapellis) covering a layer of parallel cylindrical hyphae 3–9 µm wide (mediopellis); subpellis made of broader, ventricose, short septate hyphae 15–20 µm wide; strongly encrusting, brownish pigment as coarse granulations in the pileipellis, and as small plaques or clustered granules in the subpellis. Clamps present at all septa.

*Habitat and distribution:* Reported from several alpine localities in France and Switzerland above 2300 m asl and from the Swedish Lapland, in snowbeds with *Salix herbacea*, but so far only confirmed from France and Italy, always with *Salix herbacea* and occasional *Alchemilla pentaphyllea* nearby.

*Notes*: This species was described by D. Lamoure as a rare but easily diagnosable species in the field, by the blue colour of the young lamellae and basal mycelium, combined with the greyish veil that persists long at the pileus margin, and its ecology restricted to high alpine snowbeds [8]. Since then, it has been recorded only twice, from Switzerland [38] and Italy [39]. An additional collection from M. Maletti (Italy), is reported and illustrated online [40]. However, without molecular support, there is no evidence that any of these collections are conspecific with Lamoure’s taxon. None of the recent revisions of the subgenus [2,14] included this species.

To complete our survey of alpine Agaricales of the Vanoise area [40], we succeeded in finding at LY and sequencing the holotype collection of *C. caesionigrellus* (L. 71-51), here confirmed as a member of the sect. *Flexipedes* (Figure 1). The phylogenetically closest species is the non-alpine species *C. umbrinolutescens* Reumaux [2], with four SNPs + eight indels separating the two ITS type sequences. No sequence with more than 97.9% identity with that of L. 71-51 is currently (5 August 2023) available in GenBank or UNITE, suggesting that *C. caesionigrellus* is probably a rare species at the global scale. Serendipitously, A. F. deferred to J.-M. B.’s expertise on the sequence of a small alpine *Telamonia* collected under *Salix herbacea* in the Italian Alps, which did not match anything by Blast in GenBank. This sequence turned out to differ from that of L. 71-51 by a unique SNP and is thus considered here as representing the first DNA-confirmed record of this species since its original description (Figure 1). This unexpected finding allows us to update Lamoure’s original diagnosis and to reliably illustrate this species for the first time.

***Cortinarius flexipes var. brachyspermus*** (Lamoure) Bellanger, **comb. nov**. MycoBank MB849754

Basionym: *Cortinarius paleifer* var. *brachyspermus* Lamoure, arctic and alpine Mycology, II: 258 (1987)

*Material studied*: FINLAND, Lapland, amongst *Polytrichum norvegicum* and *Sibbaldia procumbens* under *Salix herbacea*, 1972, D. Lamoure (LY L. 72-64, holotype). GenBank ITS OR419960.

*Notes*: The ITS sequence of the holotype of *Cortinarius paleifer* var. *brachyspermus*, which we could generate from the original material at LY, turned out to be almost identical (one SNP + two indels) to that of the neotype of *C. flexipes* (Pers.) Fr., in sect. *Flexipedes* (Figure 1). This finding supports the original treatment of the present taxon as a variety of *C. paleifer* Svrček, the latter name being considered a late synonym of *C. flexipes* by several authors [41,42]. This synonymy was recently [2] confirmed by sequencing the type collections of both taxa. As a result, and because of the morphological and ecological singularities of this alpine taxon described in detail by Lamoure [35,43], we here recombine this variety under *C. flexipes*. No recent collection or illustration of this variety is known to us.

***Thaxterogaster dovrensis*** (Brandrud) Bellanger, **comb. & stat nov.** MycoBank MB849755 (Figure 6).

*Basionym: Cortinarius argenteolilacinus var. dovrensis* Brandrud, in Brandrud et al., Mycol. Progr., 17 (12): 1346 (2018)

*Material studied*: FRANCE, Peisey-Nancroix, along the GR5 path to the lake of La Plagne, 2050 m asl, alpine meadow with *Helianthemum nummularium* and *Salix retusa*, close to *Dryas octopetala* and *Polygonum viviparum*, 25 August 2018, F. Armada (LY FA4340). GenBank ITS OR419956.

*Notes*: In their revision of sect. *Riederi*, Brandrud et al. [44] taxonomically treated *C. argenteolilacinus* M.M. Moser as an inclusive species with two varieties: var. *argenteolilacinus* for mainly nemoral-montane *Fagus* (-*Tilia*)-associated populations and a newly introduced var. *dovrensis* Brandrud for mainly northern, subarctic *Betula*-associated populations. Despite the fact that translating phylogeny into taxonomy is largely a matter of human choice, this one is a bit surprising because the two sister clades are well-supported and do not display any sign of gene flow. Moreover, as the authors rightly pointed out, they are associated by a combination of ecogeographical, morphological, and anatomical conspicuous differences that would largely support the species rank for these two sister taxa. In their comments, the authors justified their choice of the varietal rank over the subspecies rank (“due to geographical differences, these taxa might also be treated as subspecies, but we think it is wise to stick to only one infraspecific rank (subspecies is nowadays very little applied in agaric taxonomy”) [44], but never evoked a possible species rank for *dovrensis*. Most likely, their conservatism resulted from the short phylogenetic distance between the two clades: “the taxon is phylogenetically very closely related to *C. argenteolilacinus* var. *argenteolilacinus* and differs only in two to three substitution and indel positions in the ITS region”. One could argue that a number of genuine *Cortinarius* species (i.e., differing by clear and not overlapping morphological or ecological features) differ by less than three nucleotide changes, especially in the subgenus *Telamonia* [2,14]. Others may argue that phylogeny is not an alternative way but a tool to serve taxonomy, and that prioritizing nucleotide differences at a single locus over morpho-anatomical, biogeographical, or ecological stable associated features to assign a taxonomic rank is actually not so conservative.

The cases where infraspecific ranks have been proposed or maintained by sequence data are actually not so numerous in *Cortinarius* and they consisted of either distinct morphotaxa with no associated sequence difference (e.g., *C. atrovirens* subsp. *ionochlorus* (Maire) Vizzini & Gasparini [45]), or cryptic/semi-cryptic species with a limited number of nucleotide differences (e.g., *C. subturibulosus* var. *bombycinus* (Mahiques & Burguete) Suár.-Sant. & A. Ortega [23]). It should be noted that many of such cryptic/semi-cryptic phylospecies are later raised to the species rank, upon deeper sampling that unveils overlooked morpho-anatomic diagnostic features, or upon sequencing additional loci that unveils unreasonably long phylogenetic distances for two infraspecific taxa. *Cortinarius argenteolilacinus* var. *dovrensis* does not fit into any of these categories and deserves, in our opinion, the species rank.

In 2022, “Taming the beast […]” was published [1], a breakthrough paper splitting *Cortinarius* into 10 smaller genera. This new systematics is disruptive in many respects, but it is grounded in unprecedented phylogenomic data that grant it solid evolutionary bases. In this study, *C. argenteolilacinus* and related taxa so far treated in sect. *Riederi*, nest in *Thaxterogaster*, a genus previously including only sequestrate mushrooms [46] but shown to phylogenetically represent a wider and distinct lineage within *Cortinariaceae*, sister to all other newly introduced genera [1]. Without explanation but likely applying the art. 21.2 of the current version of the Code of Nomenclature (“Code Shenzhen” [47]), Liimatainen et al. [1] changed the name of the sect. *Riederi* to sect. *Riederorum*, and further introduced subgen. *Riederorum* (erroneously labelled as *T.* subgenus *Riederi* in Figure 2) to distinguish this lineage from phylogenetically and morphologically distantly related taxa from subgenera e.g., *Multiformes* and *Scauri*. The authors have proceeded to the most relevant recombinations of species, including *Thaxterogaster argenteolilacinus* (M.M. Moser) Niskanen & Liimat.). We here follow this systematics and meanwhile raise Brandrud’s taxon to the species rank it deserves according to the above-mentioned arguments.

The collection illustrated here is the first French record for this species, as well as its first published observation in the alpine zone.

## Figures and Tables

**Figure 1 jof-09-00942-f001:**
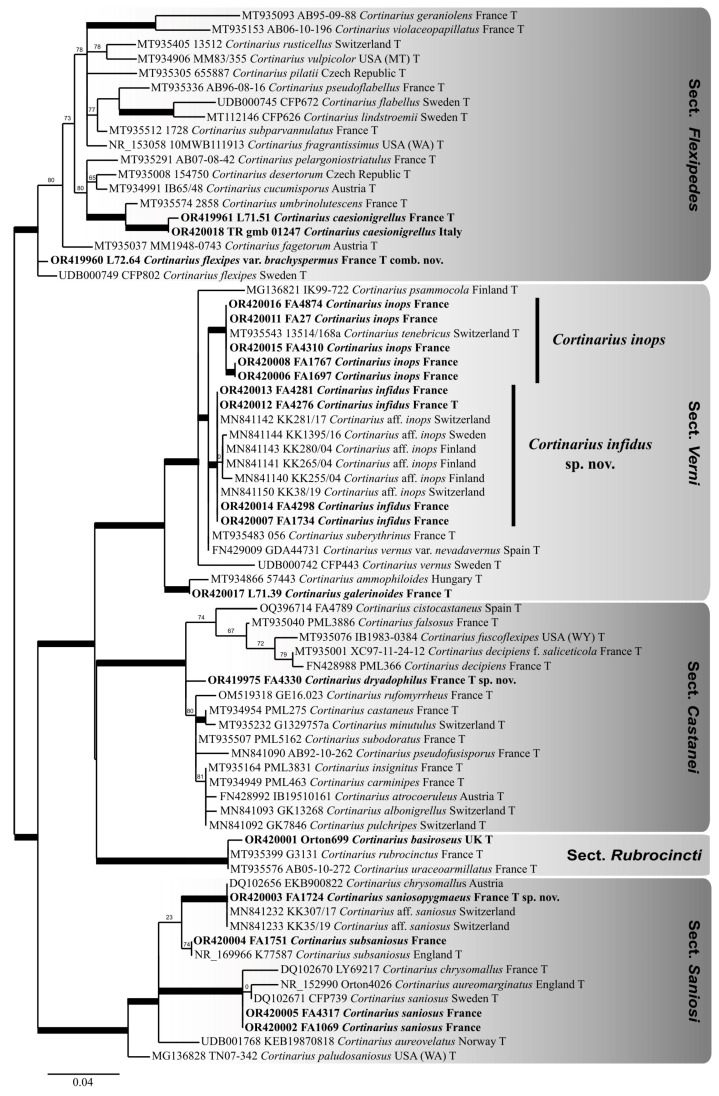
Partial midpoint rooted ML phylogenetic reconstruction of subgen. *Telamonia* from ITS sequences. In bold: sequences generated for this study.

**Figure 2 jof-09-00942-f002:**
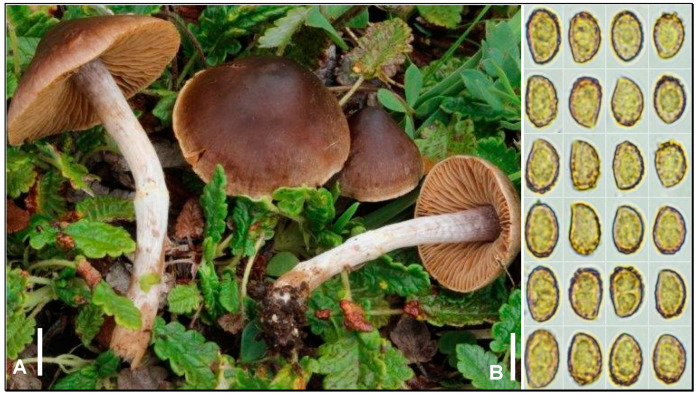
*Cortinarius dryadophilus* LY FA4330^T^. (**A**) Basidiomata in situ. (**B**) Basidiospores. Bar: 10 mm (**A**), 10 µm (**B**). Credits: F. Armada.

**Figure 3 jof-09-00942-f003:**
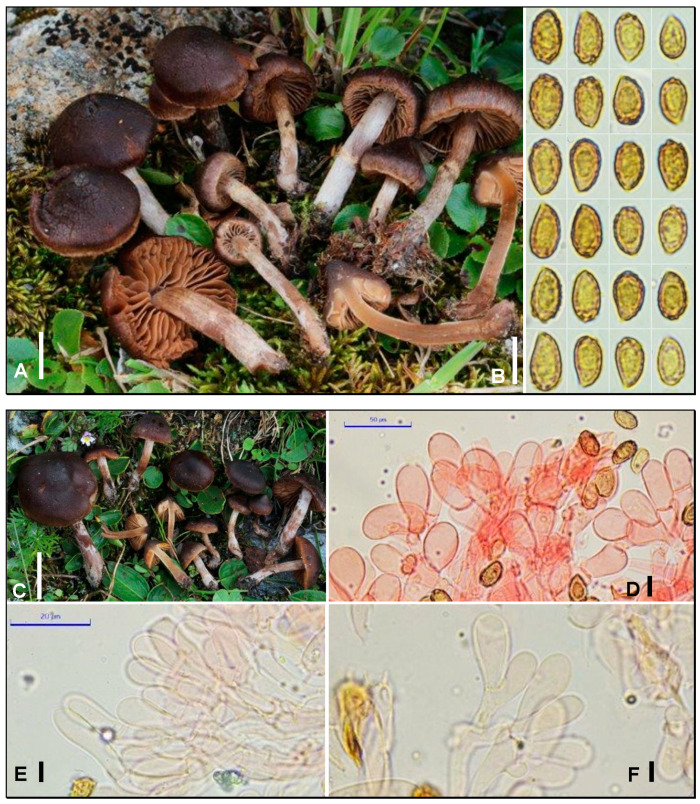
*Cortinarius infidus* (**A**,**B**,**D**) LY FA4276^T^. (**C**,**E**,**F**) FA4281. (**A**,**C**) Basidiomata in situ. (**B**) Basidiospores. (**D**–**F**) Cheilocystidia. Bar: 10 mm (**A**,**C**), 10 µm (**B**,**D**–**F**). Credits: F. Armada.

**Figure 4 jof-09-00942-f004:**
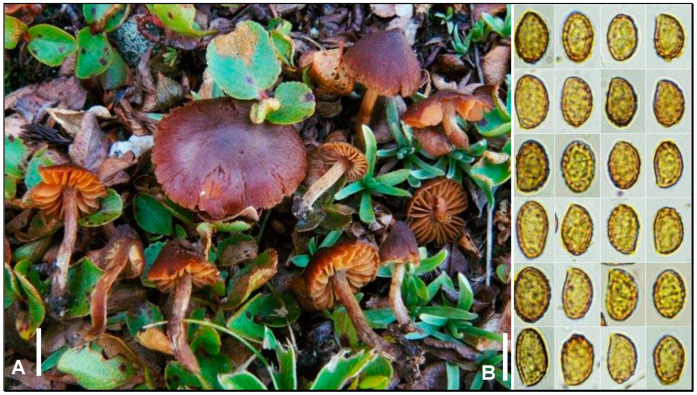
*Cortinarius saniosopygmaeus* LY FA1724^T^. (**A**) Basidiomata in situ. (**B**) Basidiospores. Bar: 10 mm (**A**), 10 µm (**B**). Credits: F. Armada.

**Figure 5 jof-09-00942-f005:**
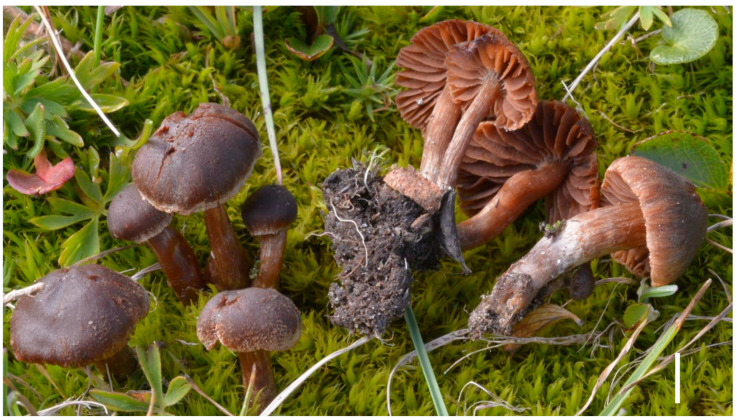
*Cortinarius caesionigrellus* TR gmb 01247, basidiomata in situ. Bar: 10 mm. Credit: A. Fellin.

**Figure 6 jof-09-00942-f006:**
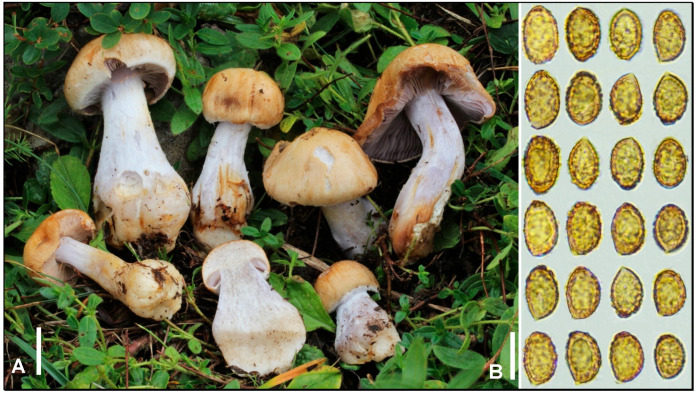
Thaxterogaster dovrensis (LY FA4340). (**A**) Basidiomata in situ. (**B**) Basidiospores. Bar: 10 mm (**A**), 10 µm (**B**). Credits: F. Armada.

## Data Availability

All DNA sequences generated in this study are deposited in GenBank. Collecting data are integrated into the MycoflAURA regional databases on fungal records (http://mycoflaura.fmbds.org, accessed on 1 August 2023) and publicly available on FongiBase (fongibase.fongifrance.fr) and the INPN website (https://inpn.mnhn.fr, accessed on 1 August 2023).

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
