# Peer review of "New Insights into Alpine Cortinariaceae (Basidiomycota): Three New Species, Two Type Revisions, and a New Record for the Alpine Zone"

_jof, 2023, doi:10.3390/jof9090942_

Round 1
Reviewer 1 Report
This study includes a lot of new information and is a great contribution to a better knowledge of the diversity and circumscription of alpine species of Cortinarius subgenus Telamonia, and Thaxerogaster.
Nomenclatural and taxonomic issues of some species names are dealt with and three new species are described as new to science.
The molecular phylogenetic assessments are based on an ITS region dataset alone but is relevant and enough to distinguish taxa on species level. The data set includes a relevant sampling of taxa to identify the sections dealt with within the genus The data set is analysed carefully using proper methods.
The manuscript is well-written. The figures are good and informative. Beside some of the diagnoses the rules of nomenclature have been followed, and the arguments and analyses are well presented. The relevant literature is properly cited.
I have no items for major revision nor any significant questions relating to the paper, just few minor comments and suggestions of improvement.
Comments:
Line 26: Suggest adding Taxonomy to the keywords
Line 99: Please specify based on what criteria these additional relevant sequences were selected? Were blast searches performed to find additional sequence data of the new described species in GenBank and the UNITE data base? If so, these databases should be cited.
Line 122: I suggest deposit an isotype at another relevant Herbarium.
Line 123: The Diagnosis should not be short descriptions characterizing the new species but rather explicitly state how the new species differ from similar phenotypic species or similar phylogenetic species. The diagnoses should tell how you properly distinguish the new species from similar and closely related taxa. Consult the code, also a diagnose is optional.
Line 145: Did Blast search in GenBank and the UNITE database not recover any further data? Maybe this should be mentioned here.
Line 201-204: See comment above on depositing an isotype and the Diagnosis.
Line 312: Why not deposit an isotype?
Author Response
Many thanks for your careful review.
Answers :
Line 26: Suggest adding Taxonomy to the keywords
- done.
Line 99: Please specify based on what criteria these additional relevant sequences were selected? Were blast searches performed to find additional sequence data of the new described species in GenBank and the UNITE data base? If so, these databases should be cited.
- We changed this sentence and replaced the last part of the sentence by "phylogenetically close published sequences from Cortinarius subgen. Telamonia, identified by BLAST searches in GenBank and UNITE".
Line 122: I suggest deposit an isotype at another relevant Herbarium.
- Unfortunately the whole collections are already registered at LY and cannot be split easily now. The Lyon University herbarium is a securized and well-managed herbarium (the 2nd of France) and we are trustful that the material is well-curated and easily accessible there currently.
Line 123: The Diagnosis should not be short descriptions characterizing the new species but rather explicitly state how the new species differ from similar phenotypic species or similar phylogenetic species. The diagnoses should tell how you properly distinguish the new species from similar and closely related taxa. Consult the code, also a diagnose is optional.
- We agree that a Diagnosis is not a description and should shortly summarize the specific features of the taxon. We think that the features we selected for C. dryadophilus (compared to C. subturibulosus but distinct by habitat and spores) and C. saniosopygmaeus (compared to C. saniosus) are conform to the expectations of the Code. We changed the diagnosis of C. infidus in order to emphasize its similarity with C. inops.
Line 145: Did Blast search in GenBank and the UNITE database not recover any further data? Maybe this should be mentioned here.
- We added "from GenBank and UNITE public databases" to the sentence. This affirmation is based on multiple investigations : BLASTs, local phylogenetical analyses, published sequences in articles, etc.
Line 201-204: See comment above on depositing an isotype and the Diagnosis.
- See answers above.
Line 312: Why not deposit an isotype?
- See answer above.
Thanks again for these helpful suggestions.
Reviewer 2 Report
Most of this reviewer's comments concern corrections of wording an grammar. This is an excellent, detailed study.
Line 23. replace at last with lastly
line 39-41. Sentence lacks a verb. Possible re-write: After Favre's era, 25 years of intensive prospections took place by Robert Kuhner and Denis Lamoure in the French alpine zone of the National Park of Vanoise.
line 47. replace was with were
line 54. insert and between Telamonia and Kokkonen: Telamonia, and Kokkonen
Line 74. Reword: Studied material: recent collections were made in Savoy, France by F. A, in Lombardy, Italy by A.F, all in alpine areas.
Line 120. is the spelling Savoy (see above) or Savoie
Line 123. spelling should be date-brown
line 129. what does "tight" mean.? this is not a familiar lamellar term. Are you referring to close which the phot suggests?
Line 137-139. what is the spore color?
Line 147. the word unless does not make sense here
Line 149. replace identifying with identify
Line 155. consists of a strongly . . . .
Line 156. associated with a handful . . .
line 157 - yet to be does not need hyphens
line 161 - replace intented with intended; replace identify with relate
line 177. , and since has been widely observed (delete comma) from the plains to the Alps under (delete under) various . . .
line 181. delete comma after [28]
line 188-189. reword: Quite surprisingly, the species most resembling C. dryadophilus macroscopically does not . . . .
line 190. delete Namely.
line 191. What is the distinctive smell?
line 233. replace dried with dry
line 247. what is the color of the spores under the microscope
line 320. ground? do you mean ground color? You mention "at first", which infers and change in color later which is not stated.
line 323. concolorous with what? - the rest of the surface
line 325. the lamellae do not look crowded in the photo, more like subdistant
line 328. background - do you mean ground color
line 331. spore color?
line 368. replace was with has been; replace so far with to date.
line 408. the lamellae look quite distant in the photo
line 409. "interspersed with short lamellae" should be moved to the end of the sentence after greyish
line 413. ground color instead of background
line 418. here you finally mention spore color!!
line 425. cylindraceous is not often seen, cylindrical is used more often
line 438. on line, should be online
line 488. replace to with by
line 505. replace in with of
many of the scientific names in the Results and Discussion are not italicized so I presume those will be changed.
I have made suggested corrections in the above box. The English is generally good but there are numerous minor corrections needed.
Author Response
Many thanks for your careful review.
Answers : All corrections listed above were accepted and reported in the revised file.
Line 23. replace at last with lastly
line 39-41. Sentence lacks a verb. Possible re-write: After Favre's era, 25 years of intensive prospections took place by Robert Kuhner and Denis Lamoure in the French alpine zone of the National Park of Vanoise.
line 47. replace was with were
line 54. insert and between Telamonia and Kokkonen: Telamonia, and Kokkonen
Line 74. Reword: Studied material: recent collections were made in Savoy, France by F. A, in Lombardy, Italy by A.F, all in alpine areas.
Line 120. is the spelling Savoy (see above) or Savoie
- We finally chose to keep the administrative French version "Savoie" as the district, more than the historical region, is considered in our paper.
Line 123. spelling should be date-brown
line 129. what does "tight" mean.? this is not a familiar lamellar term. Are you referring to close which the phot suggests?
- A badly chose word indeed, for "crowded".
Line 137-139. what is the spore color?
Line 147. the word unless does not make sense here
Line 149. replace identifying with identify
Line 155. consists of a strongly . . . .
Line 156. associated with a handful . . .
line 157 - yet to be does not need hyphens
line 161 - replace intented with intended; replace identify with relate
line 177. , and since has been widely observed (delete comma) from the plains to the Alps under (delete under) various . . .
line 181. delete comma after [28]
line 188-189. reword: Quite surprisingly, the species most resembling C. dryadophilus macroscopically does not . . . .
line 190. delete Namely.
line 191. What is the distinctive smell?
- Of orange blossoms, we added this precision.
line 233. replace dried with dry
line 247. what is the color of the spores under the microscope
- Ochre yellow, it was reported where missing for all species.
line 320. ground? do you mean ground color? You mention "at first", which infers and change in color later which is not stated.
- The sentence was re-written: "surface early dry, dark reddish-brown gradually darker at centre"
line 323. concolorous with what? - the rest of the surface
line 325. the lamellae do not look crowded in the photo, more like subdistant
- Right, "subdistant" fits well. Thank you.
line 328. background - do you mean ground color
- Yes, we will never use "background" any longer. Thank you.
line 331. spore color?
line 368. replace was with has been; replace so far with to date.
line 408. the lamellae look quite distant in the photo
- Right, the term "distant" is appopriate and added to the sentence
line 409. "interspersed with short lamellae" should be moved to the end of the sentence after greyish
line 413. ground color instead of background
line 418. here you finally mention spore color!!
- It was added were missing. Thank you for pointing that.
line 425. cylindraceous is not often seen, cylindrical is used more often
line 438. on line, should be online
line 488. replace to with by
line 505. replace in with of
many of the scientific names in the Results and Discussion are not italicized so I presume those will be changed.
- Italics and other other typographic incomes were checked and corrected as much as we could.
- Thank you again for this careful review.